# The role of formal social networks in mitigating age-related mental stress among older Nigerians living in poverty: Insights from social capital theory

formal social networks; mental stress; older adults; poverty; social capital theory

**Corresponding author:**
Sunkanmi Folorunsho;
Email: sfolorunsho2@huskers.unl.edu

Sunkanmi Folorunsho[1] , Munirat Sanmori[2] and Medinah Suleiman[3]

[1]Department of Sociology, University of Nebraska-Lincoln, Lincoln, NE, USA; [2]Department of Sociology, Georgia State University, Atlanta, GA, USA and [3]Department of Common and Islamic Law University of Ilorin, Ilorin, Nigeria

## Abstract

As Nigeria's aging population increases, older adults living in poverty face growing threats to their psychological well-being. This study examines the role of formal social networks such as government programs, non-governmental organizations and faith-based initiatives in alleviating mental stress, defined as persistent psychological distress characterized by anxiety, loneliness and emotional strain, distinct from clinically diagnosed mental illness. Using Social Capital Theory as a guiding framework, the review explores how bonding, bridging and linking social capital influence the ability of formal networks to reduce financial insecurity, social isolation and health-related vulnerabilities. Traditional family caregiving structures are weakening due to rapid urbanization and economic pressures, leaving many older Nigerians unsupported. Although formal initiatives like the National Social Safety Nets Project exist, their effectiveness is limited by delayed disbursements, poor coordination and cultural stigma surrounding mental health. Strengthening the National Senior Citizens Centre as a coordinating body, expanding culturally relevant community-based care and integrating informal support systems are identified as crucial steps forward. Without such reforms, the continued neglect of this population risks worsening mental health outcomes, straining public health resources, and undermining intergenerational solidarity. This review offers actionable insights for improving older adult-care systems in Nigeria and provides guidance for other low-resource settings confronting similar demographic transitions.

## Impact statement

This study points to the critical role of formal social networks in alleviating age-related mental stress among older Nigerians living in poverty, a growing yet underserved population in Nigeria. By applying Social Capital Theory, the research underlines how bonding, bridging and linking social capital can enhance the accessibility and effectiveness of support systems. As traditional family structures erode due to urbanization and economic challenges, formal networks such as government programs, non-governmental organizations and faith-based initiatives offer significant potential to address mental stress caused by poverty, social isolation and health vulnerabilities. The study's findings have far-reaching implications for policymakers, practitioners and community leaders. Recommendations include strengthening formal support structures through increased funding, improved coordination and culturally sensitive approaches that align with Nigeria's socio-cultural norms. Integrating informal community practices, such as savings groups and traditional care systems, into formal frameworks can improve accessibility and sustainability. Expanding pension coverage and healthcare services, particularly in rural areas and leveraging trusted institutions like faith-based organizations can reduce stigma and enhance trust in formal mental health interventions. Beyond addressing immediate needs, these measures can foster economic resilience, improve mental health outcomes and promote social inclusion for older Nigerians. The study contributes to a broader understanding of the intersection between poverty, aging and mental health in low-resource settings, offering important insights for other developing nations grappling with similar challenges. By advocating for integrated and culturally grounded solutions, the research aims to inform policies that prioritize the dignity and well-being of older adults, ensuring their inclusion in Nigeria's socio-economic development agenda.

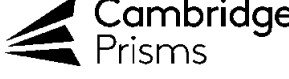

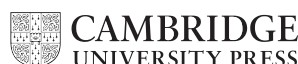

## Introduction

Nigeria's aging population is expanding rapidly, yet the country's social and economic infrastructure remains ill-equipped to meet the needs of this growing demographic (Mbam et al., 2022). The United Nations (2020) defines older adults as individuals aged 60 years and above, and in 2020, an estimated 9–10 million Nigerians fell within this category. Projections indicate that this number will double by 2050 (World Bank, 2023). Despite this rapid growth, existing policies and support systems are inadequate, raising concerns about the country's ability to provide adequate care and ensure the well-being of older adults (Oyinlola, 2024). Without significant improvements in social welfare programs, Nigeria faces an impending crisis in older adult care.

The connection between population aging and widespread poverty has given rise to a highly at-risk group – older adults struggling with economic hardship and social isolation (Mahmoud et al., 2023; Oni-Eseleh and Badaiki, 2024). These challenges contribute to mental stress, a growing but often overlooked public health concern. Mental stress encompasses persistent psychological distress, including anxiety, social isolation and depression, often triggered by financial instability, deteriorating health and weakened social support systems (Donovan and Blazer, 2020). While mental stress is distinct from clinically diagnosed psychiatric conditions such as major depressive disorder or generalized anxiety disorder, it significantly impacts quality of life and, if left unaddressed, may escalate into severe mental health conditions (Crielaard et al., 2021). Despite its prevalence, mental stress among older Nigerians remains largely understudied, as research has traditionally focused on clinically diagnosed psychiatric disorders (Ekoh et al., 2020).

Historically, Nigeria's extended family networks served as a crucial safety net for older adults, ensuring financial and emotional support through intergenerational cohabitation (Salami and Okunade, 2020). However, rapid socioeconomic changes are eroding these traditional support systems. Urbanization and migration patterns particularly the increasing trend of younger generations relocating to urban centers or emigrating for economic opportunities, a phenomenon locally known as "*Japa*" (Iwuagwu et al., 2022) have disrupted familial structures. Studies in southern Nigeria suggest that when adult children migrate for employment, their older parents experience heightened loneliness and emotional neglect (Akosile et al., 2023; Folorunsho and Okyere, 2025). Additionally, economic pressures have diminished the ability of younger family members to provide consistent caregiving, as financial survival often takes precedence over older adult care responsibilities (Ezulike et al., 2024).

Intensifying these challenges, Nigeria lacks adequate formal support structures to address the needs of its aging population (Mobolaji, 2024). The country's social security system primarily benefits retirees from the formal workforce, excluding the majority of older individuals who spent their working years in informal sectors such as farming, petty trade, and artisanal work (Oyinlola, 2024). Additionally, geriatric healthcare and mental health services remain severely underfunded and largely inaccessible, particularly in rural areas (Mahmoud et al., 2023). The COVID-19 pandemic increased these vulnerabilities, exposing the fragility of Nigeria's older adult-care systems. During lockdowns, many older adults experienced disruptions in access to health services, medication and social support. Restrictions on movement and the prioritization of pandemic-related care left chronically ill and economically disadvantaged older persons with minimal assistance, contributing to increased psychological distress (Ekoh et al., 2020; Ikeorji et al., 2024).

The pandemic also deepened isolation, particularly for older adults without active family networks, as communal gatherings and religious activities which are key sources of social engagement were suspended (Mahmoud et al., 2023). As a result, older Nigerians, especially those living in poverty, face multiple intersecting challenges, including chronic illnesses with limited healthcare access, financial insecurity due to inadequate pension coverage, and intensified social isolation resulting from both structural neglect and pandemic-related disruptions (Ekoh et al., 2020). Empirical studies emphasize a strong correlation between financial hardship and late-life depression. In Oyo State, older adults experiencing economic strain were found to be over four times more likely to suffer from depression (Baiyewu et al., 2015). Similarly, in Cross River State, nearly 45% of surveyed older adults reported depressive symptoms, with social disconnection being a primary contributing factor (Akosile et al., 2023).

This literature examines the role of formal social networks in alleviating mental stress among older Nigerians by evaluating their impact, integration with informal caregiving structures and policy implications. While aging research has predominantly focused on high-income nations, there remains a significant gap in understanding the mental health challenges faced by older adults in sub-Saharan Africa (UN, 2020). By focusing on Nigeria, this review contributes to the growing body of research on aging in low-income countries. Specifically, it assesses how formal support systems, including government programs, non-governmental organizations (NGOs), and faith-based initiatives, mitigate financial strain, social isolation and psychological distress among older adults which are contributory indices of mental stress. Furthermore, it examines the extent to which these formal networks complement or conflict with traditional caregiving structures and identifies barriers to access. Lastly, this review offers evidence-based policy recommendations to enhance the sustainability and effectiveness of older adult support systems. The findings are particularly relevant to policymakers, as strengthening formal social networks can improve mental well-being and ensure dignity for Nigeria's aging population in the face of rapid demographic changes.

## Age-related mental stress in Nigeria

Research on aging in Nigeria consistently draws attention to the mental health challenges faced by older adults, often characterized by heightened stress, depression and anxiety linked to socioeconomic hardship. While studies may not always explicitly use the term *mental stress*, they frequently document the underlying factors contributing to psychological distress. A recurring theme in this literature is the strong correlation between poverty and late-life depression. Baiyewu et al. (2015) reported that older adults experiencing financial hardship in Oyo State were over four times more likely to suffer from depression. This finding underlines the significant role of economic insecurity, particularly in the absence of pensions or social safety nets, as a key stressor in later life. Similarly, Abdullateef et al. (2018) found that in rural northern Nigeria, unstable income and poverty were closely associated with symptoms of mental distress among older adults. These studies align with broader evidence suggesting that financial insecurity in later life is not merely a material concern but also an emotional burden, as it undermines an individual's sense of stability and control (Rahman and Steeb, 2024).

Beyond financial difficulties, social isolation has emerged as another critical predictor of mental stress among older Nigerians. Tuki (2025) found that a large number of older Nigerians rely on family support for survival, yet increasing economic pressures have led to reduced financial assistance. As family sizes shrink and migration patterns shift younger generations to urban centers, many older adults experience limited social engagement and diminished familial support. Akosile et al. (2024) reported that in Cross River State, 45.5% of older adults exhibited significant depressive symptoms, with insufficient social connections identified as a primary contributing factor. These findings are consistent with global literature on aging, which recognizes social isolation and loneliness as major risks for poor mental health in later life. However, in Nigeria, these issues are exacerbated by cultural expectations that family members will provide care and companionship in old age. When these expectations go unmet, older adults may experience a heightened sense of abandonment, exacerbating psychological distress. A qualitative study in Enugu State found that older adults without active family engagement reported feeling forgotten and irrelevant in their communities, contributing to symptoms of anxiety and emotional distress (Okoye, 2024).

### Effectiveness of formal social networks in alleviating mental stress

Formal social networks encompass structured support mechanisms facilitated by governmental institutions, non-governmental organizations (NGOs), and faith-based initiatives that extend beyond immediate family structures (Collinson, 2011). In Nigeria, these networks serve as critical intervention points for older adults, particularly those experiencing financial hardship. While their effectiveness in mitigating mental stress remains understudied, preliminary evidence suggests that these networks alleviate psychological distress among older Nigerians living in poverty (Mbam et al., 2022).

At the governmental level, the National Social Safety Nets Project (NASSP) is a significant policy initiative aimed at supporting vulnerable households, including older adults. Through the National Cash Transfer Office, NASSP administers unconditional cash transfers to eligible recipients, reducing financial strain (National Social Safety Nets Project, 2023). Although direct empirical evaluations of NASSP's impact on mental health are limited, studies from similar contexts indicate that cash transfer programs improve psychological well-being by alleviating financial stress (Fenny, 2017). Anecdotal reports from Nigeria suggest that state-level programs such as the "Owo Arugbo" elderly grant in Ekiti State provide not only economic relief but also a sense of dignity and social belonging (The Sun, 2023).

Healthcare-focused formal networks also play a crucial role in addressing mental health concerns among older Nigerians. The National Health Insurance Scheme (NHIS) is theoretically designed to provide healthcare coverage; however, its reach remains inadequate, particularly for those outside formal employment (Adesanya et al., 2023). To address these gaps, alternative models such as community-based health insurance schemes (CBHIS) and NGO-led clinics have emerged. For instance, mobile health clinics, such as Wellness-on-Wheels, originally established to combat infectious diseases, now offer general health outreach, including mental health screenings and counseling for older adults (Shepard et al., 2023). Preliminary evaluations indicate that older Nigerians receiving regular visits from health outreach teams report lower levels of anxiety and improved emotional well-being, though further research is needed (Mahmoud et al., 2023).

NGOs complement governmental efforts, particularly in regions where formal state support is insufficient. Organizations such as HelpAge Nigeria implement community-based healthcare programs, older persons' clubs, and livelihood support initiatives that address both medical and social needs (Mbam et al., 2022). Participation in these programs has been linked to enhanced well-being by mitigating loneliness and increasing access to essential services (Ubaka et al., 2018). Another significant initiative is the Association for Family and Reproductive Health (ARFH), which operates adult daycare centers in urban areas, providing mental stimulation, social engagement and respite from social isolation (Eze et al., 2024). Faith-based organizations also contribute meaningfully to older adult support services. The Christian Rural and Urban Development Association of Nigeria (CRUDAN) has organized free medical eye camps for seniors, while Jama'atu Nasril Islam (JNI) facilitates food distribution to impoverished older adults, particularly during Ramadan (Klinken, 2016). These initiatives primarily address physical needs while also fostering social inclusion and alleviating stress.

Comparative assessments of formal support networks on mental health in Nigeria remain limited. However, evidence from other African countries stresses the potential benefits of well-structured support systems. In Ghana, participation in the NHIS has significantly improved healthcare access for older adults and reduced financial stress related to medical expenses (Fenny, 2017). Similarly, South Africa's Old Age Grant has lowered poverty rates and strengthened intergenerational financial support, reinforcing older adults' sense of purpose and social stability (Thovoethin and Ewalefoh, 2018).

Despite the existence of formal support systems such as NHIS and pension programs in Nigeria, their reach and efficacy remain limited. Many older adults, particularly those in informal employment sectors or rural communities, lack access to these benefits, leaving them financially and socially vulnerable. Expanding health insurance coverage and strengthening pension schemes could provide much-needed relief, as evidenced by successful models in Ghana and South Africa. Additionally, implementing CBHIS, akin to Rwanda's Mutuelles de Santé, could enhance healthcare accessibility for older adults in underserved communities (Shepard et al., 2023). Apart from governmental initiatives, collaborative efforts between NGOs, religious organizations and community-based groups are essential for sustainable older adult support systems. Successful models in West Africa, such as HelpAge International, have demonstrated the benefits of integrating healthcare, social engagement and financial assistance programs for older populations (Baillie et al., 2015).

Table 1 presents a comparative analysis of formal social support networks in Nigeria and their counterparts in other African and international contexts. While Nigeria has existing initiatives, the implementation of these programs remains fragmented and often inaccessible to a large portion of older adults, particularly those outside the formal employment sector.

### Integration of formal and informal networks

Older adults in Nigeria rely on both informal and formal support structures to navigate daily life. Traditionally, families have served as primary caregivers, but economic changes, urban migration and weakening extended family networks have

**Table 1.** Selected formal social support networks in Nigeria compared to other African countries

| Formal social network in Nigeria | Examples from other African countries |
| --- | --- |
| National Social Insurance Trust Fund (NSITF) | Similar pension schemes (e.g., *Old Age Grant*) in countries like South Africa have helped reduce poverty in older adults (Thovoethin and Ewalefoh, 2018). |
| National Health Insurance Scheme (NHIS) | The NHIS in Ghana has been remarkable and it is a viable tailored service for older adults (Fenny, 2017). |
| Pension Scheme | The South African Old Age Pension has significantly reduced poverty in older adults (Thovoethin and Ewalefoh, 2018). |
| Community-Based Health Insurance Schemes (CBHIS) | Community health insurance (*Mutuelles de Santé*) in Rwanda has been effective in rural health access (Shepard et al., 2023). |
| Non-Governmental Organizations (NGOs) | HelpAge International has been active in providing support for older adults in West Africa |
| Faith-Based Organizations | Faith-based services in countries like Uganda (*The Butabika-East London Link* and *Joy Initiatives Uganda)* offer vital mental health and social support (Baillie et al., 2015). |
| Conditional Cash Transfer (CCT) Programs | Brazil's *Bolsa Família* has successfully reduced poverty with CCT programs (Shei et al., 2014). |
| Old People's Homes and Day Care Centers | Malaysia's senior care centers like Sunway Sanctuary and GreenAcres are providing comprehensive care services to older adults (Md Isa et al., 2022) |
| Legal Aid Council of Nigeria | The UK's Legal Aid Agency provides similar support for vulnerable older populations (Creutzfeldt and Sechi, 2021). |
| Integrated Social Protection Programs | Mexico's social protection programs (e.g., *Programa Pensión para el Bienestar de las Personas Adultas Mayore*) have reduced vulnerability of older adults through integrated services. |

strained these arrangements (Ojagbemi et al., 2020). As formal support networks bridge these gaps, their effectiveness depends on how well they integrate with existing informal caregiving structures. Successful models reinforce rather than displace traditional caregiving networks. For example, targeted cash transfer programs not only provide direct financial relief but also reduce the economic burden on caregivers. Family members and community leaders often act as facilitators, assisting older adults in enrolling in government programs or accessing NGO-led initiatives. Studies indicate that older adults with active family support are more likely to participate in formal services, achieving better health outcomes (Subu et al., 2022). However, socially isolated seniors may struggle to access programs due to a lack of awareness or distrust of formal institutions, prompting NGOs to establish community-based referral systems leveraging trusted local figures (Mentally Aware Nigeria Initiative [MANI], 2023). Some initiatives have successfully integrated formal and informal support systems. In Anambra State, the "Care for the Elderly" initiative deploys healthcare workers to conduct home visits, ensuring older adults receive medical attention while remaining in familiar environments (Tanyi et al., 2018). This model acknowledges older adults' preference for home-based care over institutionalized settings, demonstrating the feasibility of harmonizing formal and informal older adult care systems (HelpAge Nigeria, 2022).

## Social capital theory

This review employs Social Capital Theory to examine how formal social networks mitigate mental stress among older Nigerians living in poverty. Social capital refers to the resources embedded in social relationships that provide individuals with emotional support, financial assistance and access to essential services (Bourdieu, 1986; Putnam, 2001). The theory is typically categorized into bonding, bridging, and linking social capital, each playing a distinct role in improving well-being and reducing psychological distress.

## Bonding social capital

Bonding social capital refers to the resources and support exchanged within close-knit networks, such as families, neighbors and community groups, which rely on strong interpersonal trust and mutual obligations (Putnam, 2001). In Nigeria, these relationships form the backbone of older adult care, as extended family structures have historically provided financial assistance, daily caregiving and emotional support for older adults. The traditional Yoruba concept of *omoluwabi*, which emphasizes respect and duty toward older adults, reinforces caregiving expectations within families. Similarly, among the Igbo, the age-grade system (*otu ogbo*) fosters communal responsibility for older adult welfare, ensuring that older adults without immediate family support receive assistance from peers (Mba et al., 2023). Outside familial structures, indigenous self-help associations continue to play an essential role in sustaining older adult support. *Osusu* and *Adashe*, traditional rotating savings and credit schemes, provide financial buffers for older adults, particularly those in informal employment sectors (Omari et al., 2024). Women's cooperatives in the Middle Belt and southeastern Nigeria organize mutual aid funds that cover medical expenses for aging members, while *Nze na Ozo* societies among the Igbo ensure that older members receive financial and social support, especially during health crises (Mba et al., 2023). These informal structures reduce dependence on formal social welfare schemes, particularly in rural communities with limited government intervention. However, economic migration and urbanization have significantly weakened these traditional safety nets. In response, faith-based older adult support groups have emerged, such as the Catholic Women's Organization (CWO) and Muslim Sisters Organization (MSO), which provide financial assistance and visitation services for abandoned older adults. Integrating formal older adult-care policies with these existing traditional networks can enhance support for Nigeria's aging population.

## Bridging social capital

Bridging social capital includes the connections that link individuals to broader networks and resources outside their immediate social circles, facilitating access to new opportunities and services (Putnam, 2001). In Nigeria, bridging social capital plays a critical role in older adult support by connecting older adults to religious institutions, NGOs, community-driven healthcare initiatives and social inclusion programs that enhance their well-being. Religious

organizations serve as a major source of bridging social capital, filling gaps where formal state support is limited. The Justice, Development and Peace Commission (JDPC) of the Catholic Church provides essential welfare services, including food distribution, free health screenings and social gatherings for older parishioners, thereby addressing both financial hardship and loneliness (Justice, Development and Peace Commission, 2019). Similarly, mosque-based charities, such as Zakkat Foundations in northern Nigeria, distribute stipends and medical assistance to older adults, particularly widows and those without family caregivers.

Community-driven healthcare programs further illustrate the role of bridging social capital in improving health outcomes among older Nigerians. The Wellness-on-Wheels initiative, initially launched to combat communicable diseases, has been expanded to offer mobile hypertension and diabetes screenings for older adults, ensuring access to medical attention in underserved communities (Ezulike et al., 2024). In Lagos, the Elderly Health Fund (EHF), a public-private partnership, subsidizes medical expenses for low-income seniors who would otherwise be unable to afford out-of-pocket payments.

Apart from financial and medical support, bridging social capital fosters social inclusion through structured community engagement programs. In Enugu and Oyo states, Community Older adults' Forums have been established to offer recreational activities, peer counseling and skill-building workshops for older adults, helping to maintain their cognitive and emotional well-being. Additionally, local governments in some states have partnered with NGOs to organize intergenerational mentorship programs, where older adults share knowledge and skills with younger community members, promoting social integration and a sense of purpose. While these initiatives have proven effective, their sustainability depends on continued investment and collaboration between government agencies, religious institutions and civil society organizations. Expanding these programs through strategic funding and policy support could further enhance their reach, ensuring that more older Nigerians benefit from the opportunities provided by bridging social capital.

## Linking social capital

Linking social capital refers to the connections between individuals and formal institutions, including government welfare programs, healthcare systems and legal aid. In Nigeria, this dimension is critical for ensuring that older adults, particularly those in poverty, can access structured social protection mechanisms. Government-led cash transfer programs serve as primary linking mechanisms. The NASSP provides stipends to vulnerable households, including older Nigerians, to mitigate economic distress (National Social Safety Nets Project, 2023). Similarly, the *Owo Arugbo* scheme in Ekiti State offers targeted financial relief to older individuals with no regular income (The Sun, 2023).

Advocacy organizations have also played a significant role in strengthening and linking social capital. HelpAge Nigeria has successfully lobbied for pension reforms, increased healthcare funding and legal protections for older citizens. Additionally, the Coalition for the Rights of Older Persons in Nigeria (CROP-N) has pushed for the implementation of the National Senior Citizens Act, which aims to establish dedicated older adult-care services across the country. Legal aid organizations such as Human Rights Law Service (HURILAWS) provide free legal assistance to older Nigerians facing pension disputes or financial exploitation, ensuring that they receive the benefits to which they are entitled.

Healthcare access remains a critical area where linking social capital can be strengthened. The National Health Insurance Scheme (NHIS) includes provisions for older adults, but coverage is limited, particularly for those outside formal employment sectors. In response, private-sector collaborations have emerged, such as Hygeia's Elderly Care Plan, a subsidized insurance package specifically designed for retirees. Scaling such models through public-private partnerships could significantly improve healthcare access for Nigeria's aging population.

## Challenges and opportunities in strengthening formal networks

The expansion of formal social networks for older Nigerians faces significant financial, policy, cultural, governance and human resource challenges. Addressing these barriers presents opportunities to enhance older adult support systems through innovative reforms. One of the most pressing challenges is financial sustainability, as many government initiatives, including cash transfer programs, experience budget shortfalls or rely on unstable donor funding. Programs such as the NASSP have reported delayed disbursements due to financial constraints, reducing their impact on vulnerable older adults (World Bank, 2024). Establishing a dedicated Older Adult Care Fund, supported through tax levies or redirected subsidies, could provide stable financing. Public-private partnerships (PPPs) also offer an opportunity for sustainable funding, with private sector actors such as telecom and financial institutions contributing to mobile health clinics and senior welfare centers as part of corporate social responsibility initiatives.

Fragmentation and poor coordination among social welfare agencies further weaken the effectiveness of formal networks. Different ministries manage parallel programs with little collaboration, leading to inefficiencies and service gaps (Aregbeshola, 2021). Strengthening the National Senior Citizens Centre (NSCC) as a coordinating body could harmonize services and streamline older adult support efforts. A national database of older adult support programs would help eliminate redundancy and ensure resources are efficiently allocated (Aregbeshola, 2021). Lagos State has piloted integrated senior welfare centers that combine healthcare, financial assistance and social services under one roof, offering a scalable model for nationwide implementation. Establishing an Older Persons' Council composed of government officials, NGOs and community representatives could further improve participatory planning and oversight of older adult-care policies.

Cultural barriers also limit participation in formal older adult support programs. Many older Nigerians perceive mental distress as a spiritual issue rather than a medical condition, discouraging them from seeking professional mental health services. Additionally, financial aid is sometimes viewed as a sign of family neglect, leading some older adults to reject formal support. To address these concerns, culturally sensitive outreach strategies are needed. Religious and traditional leaders should be engaged to promote the benefits of older adult support programs, as faith-based institutions already provide food aid, health services and emotional support to older adults. Community-based structures, such as *Osusu* and age-grade associations, could be integrated into formal networks to improve acceptance. Recognizing and funding existing older councils in local communities would further enhance accessibility and trust in formal programs.

Governance and accountability challenges have also hindered the success of older adult support programs. Reports of corruption and mismanagement in pension and welfare schemes have eroded public trust, making it essential to strengthen oversight mechanisms. Expanding biometric smart cards and mobile payment systems, already used in Nigeria's contributory pension scheme, to non-contributory pensions and cash transfers could improve efficiency and transparency (Aregbeshola, 2021). Community watchdog groups and independent NGOs could play a role in monitoring service delivery and ensuring that funds reach their intended beneficiaries. Establishing whistleblower protections and a grievance redress system specifically for older adults would further enhance accountability. A digital dashboard, similar to the pension transparency system used in Lagos, could provide real-time tracking of older adult support services, increasing confidence in the system.

The shortage of trained personnel is another critical challenge limiting the reach and effectiveness of formal older adult-care services. Nigeria lacks sufficient geriatricians, mental health professionals and social workers, making it difficult to provide specialized care for older adults. Many primary healthcare centers do not have personnel trained to manage aging-related health conditions, particularly mental health and chronic illnesses. A practical solution is task-shifting, where community health workers (CHWs) and nurses receive specialized geriatric care training (Ikeorji et al., 2024). This approach has been successfully implemented in Rwanda and could be adapted to Nigeria. Engaging older adults as peer supporters could also help address emotional well-being, following models used in South Africa's senior companion program. Additionally, incorporating geriatric care into medical and social work training curricula would help build long-term expertise in older adult care. Expanding partnerships with faith-based hospitals and mission clinics, which already provide a significant portion of Nigeria's healthcare services, could further strengthen geriatric mental health programs (Ikeorji et al., 2024).

## Policy recommendations for strengthening formal social networks for older adults in Nigeria

To enhance formal social networks for older Nigerians, several targeted policy measures should be implemented. Establishing community-based older adult support centers at the local level would improve access to pensions, healthcare check-ups and mental health screenings. These centers should be co-funded by the government, NGOs, and private partners to ensure sustainability. Leveraging existing primary healthcare clinics to provide dedicated older adult-care services on specific days could expand service coverage without the need for significant new infrastructure investment.

Expanding non-contributory pensions and cash transfer programs would reduce financial insecurity among older adults. A universal or means-tested pension scheme, starting with individuals aged 70 and above, could be gradually expanded. Strengthening NASSP to ensure timely disbursements and integrating additional services such as health check-ups and psychosocial counseling into payment collection processes would enhance its effectiveness.

Integrating geriatric and mental health services into primary healthcare would improve healthcare accessibility for older Nigerians (Ikeorji et al., 2024). Training CHWs to identify and manage age-related depression and anxiety could help bridge the gap in specialized geriatric care. Expanding mobile health clinics, modeled after the Wellness-on-Wheels initiative, would provide routine screenings, mental health support and medication distribution to underserved areas. Introducing a subsidized older adult-care package under the NHIS would improve affordability and encourage participation.

Traditional and religious institutions should be leveraged to increase older adult-care accessibility. Community leaders can help identify at-risk seniors and facilitate enrollment in social programs, while age-grade associations could receive small grants to strengthen their support for older members. Religious organizations should be encouraged to establish Elder Fellowship Groups that provide social engagement, health education and access to welfare services. Training pastors and imams to recognize mental health symptoms and provide referrals would further bridge the gap between spiritual and professional support.

Public awareness campaigns are necessary to reduce stigma around mental health and encourage help-seeking behavior. Nationwide campaigns using radio, television, community dramas and religious platforms should emphasize that depression and anxiety are treatable medical conditions. Testimonies from respected older adults and religious figures could help normalize mental health interventions. Establishing "Elder Mental Health Ambassadors" in local communities would promote available services and guide seniors toward appropriate support systems.

Strengthening data collection and research is essential for evidence-based policymaking. Nigeria should conduct regular aging surveys to track mental health, living conditions and social support systems among older adults. Disaggregated data would enable targeted interventions for the most vulnerable groups. Additionally, evaluating pilot initiatives such as peer support groups and cash-plus-care models would provide insights into their effectiveness. Collaborations with universities and international research institutions could help build a robust evidence base for aging policies.

Ensuring transparency and accountability in social programs is critical for maintaining public trust. A monitoring and evaluation framework should be established to track service reach, beneficiary satisfaction and improvements in mental well-being. Independent civil society groups should be involved in tracking pension disbursements and older adult-care services. A digital dashboard, similar to Lagos State's pension transparency system, could provide real-time updates for beneficiaries. Establishing whistleblower channels and a grievance redress mechanism would allow older adults to report service failures and receive timely assistance.

## Limitation and future research

Some limitations in the present review should be noted. The first concerns the reliance on secondary data rather than primary empirical research. Although the study offers an integrative synthesis of available programs and policies, it does not present first-hand accounts of older adults who are directly affected by poverty and mental stress. While the review draws from documented interventions and government reports, the absence of primary fieldwork limits the ability to fully capture the lived realities and subjective meanings older adults attach to formal support systems. Future research should include qualitative interviews or ethnographic methods to gather narratives from older Nigerians, especially those in rural or marginalized settings, to provide richer, more grounded perspectives.

Another limitation is that the review, while drawing upon Social Capital Theory, did not empirically measure bonding, bridging, or

linking social capital across different interventions. Although these categories helped conceptualize the role of formal social networks, their actual influence on mental stress reduction remains speculative. For instance, while religious networks were identified as sources of bridging capital, there is little evidence on how sustained participation in such networks affects long-term mental well-being. Future studies might explore how the frequency of interaction with faith-based institutions or perceptions of trust in state-led programs mediate the relationship between social capital and psychological outcomes.

A third limitation relates to the demographic and contextual scope of the literature reviewed. The analysis focused primarily on older adults in urban and semi-urban settings, with less attention to those in remote or conflict-affected regions, where access to formal support is more limited and where cultural interpretations of mental stress may differ. Future research should investigate how regional disparities, language diversity and local governance structures influence the uptake and outcomes of formal support programs. Expanding the geographic and cultural scope of research would strengthen the generalizability of findings and inform more regionally appropriate interventions.

The review also did not sufficiently disaggregate findings by gender or disability status. While older adults were examined as a collective demographic, evidence suggests that older women – particularly widows – and those living with physical or cognitive impairments may face distinct barriers in accessing formal social support. These populations may be more likely to be socially isolated, less economically independent and more vulnerable to institutional neglect. Future research should adopt an intersectional lens to explore how gender, disability, and socio-cultural status intersect to shape the accessibility and effectiveness of formal support systems for mental stress reduction.

Additionally, the review included limited discussion on how older adults navigate potential conflicts between formal interventions and cultural norms surrounding filial care. In some communities, receiving assistance from external agencies may be perceived as a sign of family failure or neglect, leading older individuals to resist formal support even when it is available. Future studies could investigate how cultural expectations around dignity, reciprocity and dependence influence older adults' engagement with formal programs. Such insights would be especially valuable for designing interventions that align with local values while still meeting the psychosocial needs of aging populations.

Another limitation is the lack of systematic program evaluations in the reviewed literature. While initiatives like the National Social Safety Nets Project and various NGO-led outreach efforts were discussed, few have been independently assessed for impact, scalability, or sustainability. There is a need for longitudinal studies that evaluate not just program outputs but also long-term effects on older adults' mental health, quality of life and social inclusion. These studies should consider using mixed-methods approaches to link quantitative outcome measures with qualitative insights into user satisfaction and lived experiences.

Finally, Nigeria currently lacks a national longitudinal aging survey that includes validated measures of mental stress, health service use and social support. The absence of such data creates significant gaps in national planning and obscures the full scale of older adults' psychosocial needs. Future research should prioritize the development of aging-specific data systems that are disaggregated by age, gender, location and socioeconomic status. Doing so would provide an empirical foundation for targeted policy action and allow for better monitoring and evaluation of support programs.

## Conclusion

The evolving landscape of aging in Nigeria accentuates the critical role of formal social networks in mitigating late-life mental stress. While traditional family-based support structures have historically provided financial and emotional security, shifts in economic patterns, urbanization and migration have increasingly strained these informal caregiving systems. In response, government programs, NGOs, and religious organizations have stepped in to bridge these gaps, offering financial aid, healthcare services and structured social engagement opportunities. The findings presented in this review suggest that targeted policy interventions can enhance the efficacy of these formal support systems. Expanding noncontributory pensions, streamlining access to healthcare through community-based outreach, and strengthening collaborations between government agencies and local organizations can improve service delivery and participation. Moreover, integrating culturally grounded interventions, such as leveraging religious institutions and indigenous community networks, can increase trust and accessibility among older Nigerians, many of whom remain skeptical of formal assistance.

The broader policy implications are clear. As Nigeria's aging population continues to grow, a failure to adapt its social infrastructure will only deepen existing vulnerabilities, exacerbating financial hardship, social isolation and mental distress among older adults. A more coordinated, multisectoral approach, grounded in sustainable financing, cross-sector collaboration, and evidence-based policy design, can ensure that formal social networks function not as isolated interventions but as integral components of a more resilient and inclusive older adult-care system. Moving forward, investing in these frameworks will be essential not only for alleviating mental stress but also for safeguarding the dignity and well-being of Nigeria's aging population.

**Open peer review.** To view the open peer review materials for this article, please visit http://doi.org/10.1017/gmh.2025.10012.

**Data availability statement.** This study did not involve primary data collection. All secondary data and references used in the study are publicly available and properly cited in the manuscript.

**Author contribution.** S.F. Conceptualized the study, developed the theoretical framework, conducted the literature review, and drafted the initial manuscript. M.S.: Contributed to data synthesis, provided critical revisions, and enhanced the analysis of formal social networks and Social Capital Theory. M.S.: Provided and assisted with manuscript editing, and ensured the inclusion of culturally relevant recommendations.

**Financial support.** No external funding was received for this study.

**Competing interests.** The authors declare no conflicts of interest related to this study.

**Ethics statement.** As this study is a perspective-based analysis, it did not require Institutional Review Board (IRB) approval. The research adheres to ethical guidelines for the use of secondary data and theoretical interpretations in academic work.

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
