## [Reviewer Report]

I am really pleased to be reading this article. The manuscript, titled “Role of Formal Social Networks in Mitigating Age-Related Mental Stress Among Older Nigerians Living in Poverty: Insights from Social Capital Theory,” is both timely and significant. Firstly, I want to commend the authors for their outstanding work and for initiating a conversation on this crucial topic regarding the mental health of older adults. However, I have some feedback:

The topic appears to be objectifying the population of older adults. Firstly, it would be helpful to clarify the author’s definition of “mental stress” and “older Nigerians” and how the author classified older Nigerians. The abstract seems somewhat disjointed. It would greatly benefit from clearly articulating the main problem that this paper aims to address and what makes the idea unique. This would help capture the issues that could influence policy directives more effectively.

In the introduction section of the manuscript, several ageist statements are made, such as ‘knotty.’ It may be beneficial to reconsider the language used to avoid any derogatory implications for the status of older adults in Nigeria. Furthermore, it would be helpful to clearly define who is considered elderly and the age classification of older adults in Nigeria. Boldly addressing the issues and problems associated with the formal provision of support for older adults in Nigeria and how this can improve their mental health would enhance the paper’s impact.

What makes this paper unique? Exploring what the existing literature tells us about formalized support and how this support translates into either improvement or counterproductivity to the well-being of older adults in Nigeria would enrich the discussion. What does the policy or human rights legislation have to offer regarding the mental health and care of older adults in Nigeria? It would also be beneficial to consider the implications of the newly established National Senior Citizens Centre.

To enhance this paper, I suggest focusing on critically conceptualizing formalized support for older adults. Additionally, considering a theoretical paradigm that moves our thinking away from Eurocentric social capitalization theory to a more Afrocentric philosophical idea about supporting the mental health and well-being of older adults in Nigeria would be thought-provoking.

If the authors can critically examine the contextual factors that either inhibit or facilitate formal support services for older adults in Nigeria, this paper will undoubtedly make a valuable contribution to the field of mental health services for older adults in Nigeria.

---

## [Reviewer Report]

The manuscript provides an examination of the challenges faced by older Nigerians in accessing formal social support. The integration of Social Capital Theory is insightful, offering a strong framework for understanding the role of formal networks in mitigating age-related stress. Below, I provide section-by-section feedback on potential areas for improvement.

Introduction

• The introduction might be strengthened by further elaborating on Social Capital Theory’s relevance to Nigerian culture specifically.

• Although a general theoretical background is offered, connecting it more directly to the cultural dimensions of Nigerian society could enhance the theoretical framework’s applicability.

• On page 1, lines 46-49, use more suitable research citations from the Nigerian setting.

• The figure on poverty rates (p. 4) is not well-integrated with the narrative text. The description does not include sufficient analysis of how the data specifically affects the aging population, which would make the data more relevant.

• The introduction mentions that “over 40% of the population lives below the poverty line,” citing the National Bureau of Statistics, but the citation (2020) is arguably outdated. If more recent data are available, they would enhance the paper’s relevance.

Contextualizing Poverty and Aging in Nigeria

• While the economic challenges are well-articulated, there is limited discussion on psychological impacts specific to poverty and aging, which would help reinforce the mental health dimension of the study.

Impact of Poverty on Older Adults' Health

• A more detailed analysis of specific barriers to healthcare (e.g., cultural stigmas around mental health) could improve the section’s relevance to the study’s focus on mental health.

• A clearer distinction between informal support insufficiencies and potential roles for formal systems would enhance the reader’s understanding.

• Page 7, line 35, write NGO in full before subsequent abbreviations.

The Role of Formal Social Networks in Supporting Older Nigerians

• The section might benefit from a more systematic assessment of Nigeria’s current social programs, specifically analyzing why these systems fall short and identifying actionable steps for improvement.

• Also, discussing potential challenges to implementing new formal social networks (e.g., political, economic) could provide a more balanced viewpoint.

• The caption and content are informative, yet some example columns lack context. Clarifying how each program (e.g., NSITF and NHIS) would specifically apply to Nigerian elder care would help ensure relevance.

• In comparing Nigeria’s social systems with those in other African countries, more specific statistics would strengthen the argument. For example, if the Old Age Grant in South Africa significantly reduced poverty in older adults, providing specific data would clarify the impact.

Social Capital Theory

• The authors could strengthen this section by explaining more precisely how each form of social capital could directly mitigate mental stress.

• More examples specific to the Nigerian context (e.g., traditional support networks) could help clarify how Social Capital Theory operates practically within Nigerian communities.

Conclusion

• Summarizing potential challenges in implementing these recommendations, along with proposed solutions, would provide a more actionable and forward-looking conclusion.

Proofread for grammatical errors. For example:

• In the abstract, the phrase “Drawing from the insights from Social Capital Theory, this study point to” contains a subject-verb agreement error. It should read “this study points to.”

• In phrases like “Drawing from the insights from Social Capital Theory,” the preposition “from” is repeated unnecessarily. It should be “Drawing on insights from Social Capital Theory.”

• Throughout the manuscript, there is inconsistent use of terms such as “elderly,” “older adults,” and “aging population.” Standardizing these terms would improve clarity. Also, using the term ‘elderly’ when describing older adults is no longer acceptable.

• There is some inconsistency in referring to the National Health Insurance Scheme as both “NHIS” and its full form in various sections. Choosing and using one term consistently after the initial mention would help with readability.

• The reference style/formatting is inconsistent.

---

## [Reviewer Report]

Thank you for the opportunity to review this manuscript. This is a well written paper. The suggestions (figure 2 - should be ‘table 2’) and description of the relevance of Social Capital Theory for this topic are great.

My main comments are:

- The study is about stress but this is never defined. Please define this.

- Traditional/historical availability of family support is mentioned repeatedly. Please provide some relevant references to backup this claim. I am not very familiar with the Nigerian context but in India the absolute availability of family support has been refuted (see Penny Vera-Sanso’s work)

- Generally a lot of statements are made without any references to support them (e.g., that adults are left to manage their conditions which leads to worsening health in Nigeria, that norms restrict women’s opportunities in northern Nigeria ) / with old references (e.g., a 2004 reference regarding rising inflation) / references to studies conducted in other countries. Please ensure that all statements are referenced appropriately.

- Some points are repeated throughout the paper, please try and streamline the paper structure and points made

---

## [Reviewer Report]

Thank you for giving me the opportunity to review this manuscript for a second time. I appreciate the authors' efforts to incorporate improvements into the manuscript. However, I feel that there are still areas where clarity and refinement would significantly enhance the work. I also suggest submitting a version of the manuscript with tracked changes, which would make it easier to follow the revisions.

Abstract

The abstract still feels somewhat fragmented. For instance, the concept of “mental stress” requires clearer definition. Is it synonymous with mental illness, or does it refer to something different? Providing a concise explanation would help the reader better understand the scope of your work. Additionally, while you have outlined key actions—such as strengthening formal structures, improving coordination, and fostering culturally sensitive approaches—the abstract lacks a clear “so what” factor. Why should the reader care? What are the consequences of inaction? Strengthening the takeaway message here would add impact.

Also, a quick note: the keywords need to follow APA formatting guidelines. You can refer to the APA resource: https://apastyle.apa.org/instructional-aids/abstract-keywords-guide.pdf

The introduction is well-written, but conceptually, I remain unclear about the core idea of “mental stress” among older adults. What makes this issue unique in the Nigerian context? Moreover, why is your commentary paper particularly significant? What should researchers and policymakers (especially those you classify as formal social networks) learn from this? Most importantly, what are the risks if this issue is ignored? Adding these dimensions would greatly strengthen the manuscript’s relevance.

Age-Related Mental Stress Among Older Adults in Nigeria: This section is somewhat confusing. It’s not entirely clear whether your goal is to highlight existing empirical evidence or to argue for increased attention to the mental stress of older adults in Nigeria. If the latter, what specific gaps did you identify in the literature? And what are the implications of these gaps for policy and care for older adults in Nigeria? Additionally, most of the studies you cited are focused on mental health broadly, rather than mental stress specifically. The studies largely center on hospital settings, with only one addressing a community-based context. For instance, you mentioned the work of Animasahun and Chapman (2017), but this was not a multi-state study. Furthermore, there is a significant body of research on older adults’ mental health during COVID-19 that you seem to have missed. Including these would provide a more comprehensive foundation for your arguments. Again, ask yourself: who were these studies investigating, where were they conducted, how were they designed, and what were their key findings? Addressing these questions would improve the manuscript’s depth.

The section on formal social networks in Nigeria had some inaccuracies that need to be addressed. For example, the Association of Reproductive and Family Health does not provide services specifically for older adults. Furthermore, how have you conceptualized “formal structures”? You mentioned landlord associations, community development groups, and the Egbe Ajo system—are these not formal structures? These associations are typically governed by constitutions and legal frameworks, and many are registered organizations in Nigeria. It’s important to note that formal structures don’t necessarily need to be government-affiliated. This distinction should be clarified. Additionally, while formal social support for older adults living with mental illness is gradually emerging, the manuscript would benefit from examples of novel social support initiatives in Nigeria. For instance, you could highlight the psychogeriatric clinic at the first geriatric center in Africa. It would also be helpful to include more concrete examples of existing mental health support activities for older adults. I feel the manuscript overemphasizes the challenges related to mental health among older adults, while overlooking the broader context. For instance, the activities of NGOs like MANI (which you mentioned) are youth-focused, and they currently lack targeted support for older adults. A more balanced discussion of the existing efforts and gaps would provide a fuller picture.

There is considerable repetition throughout the manuscript, which needs to be addressed. A shorter, more concise version with tracked changes would improve readability. While the manuscript has great potential, these corrections are necessary to strengthen its contribution to understanding the intersection of poverty and mental health crises among older adults in Nigeria.

---

## [Reviewer Report]

Minor comments:

• Some statements lack citations or appropriate citations. For example, add citations on page 28, lines 47-51.

• Be consistent with the citation format used for this manuscript. Some in-text citations missed proper citations. For example, (UN, 2020) should be (United Nations, 2020). Also, (Abubakar, 2022) had more than one author and should be cited accordingly.

• Check that you are using appropriate terms. For example, “elderly” is frowned upon nowadays as an ageist term. Yet, it is a keyword for this manuscript.

Major comments:

• The narratives in the different sections are not adequately tied back to the three aims of the paper. Also, the transition from some sections/subheadings to the next is very disconnected, detracting from its readability.

• The comparative analysis (Table 1) is underexplored and lacks critical evaluation of why Nigerian programs lag behind.

• The repetitive use of “bonding,” “bridging,” and “linking” social capital without clear differentiation may confuse non-specialist readers. In this manuscript, technical terms like “bonding social capital,” “bridging social capital,” and “linking social capital” are essential but could benefit from consistent and concise definitions to aid readability.

o For example, “Bonding social capital involves close relationships, such as family or community ties, which serve as primary support systems (Bourdieu, 1986).” Page 15, lines 5-8.

o Another example, “Bridging social capital, which links diverse groups and institutions, is particularly relevant in addressing the mental health needs of older Nigerians who face geographic isolation, financial constraints, and cultural stigma.” Page 42, lines 15-22.

• Which of the World Bank-referenced citations is this sentence referring to? “In 2020, Nigeria had approximately 9.5 million older adults, a figure projected to double by 2050 (World Bank, 2022).” You have World Bank 2020 and World Bank 2023 in the reference list.

• The paper may benefit from expanding the section about Nigeria’s NHIS in the sentence, "Comparatively, Nigeria’s NHIS covers less than 5% of the population, with minimal impact on older adults due to limited rural outreach and challenges in enrollment (Adesanya et al., 2023)." What other reasons are there for why Nigeria’s NHIS is lagging? E.g., funding constraints, corrupt practices, cultural and socioeconomic barriers, etc.

• Check the sentence "Groups like the Christian Rural and Urban Development Association of Nigeria (CRUDAN) and Jama’atu Nasril Islam (JNI) offer vital services, including food assistance, medical outreaches, and spiritual counseling (Morse et al., 2023)." Though the cited article highlights leveraging support by Faith-Based Social Groups in rural villages in Nigeria, it does not explicitly mention the groups listed in the sentence. Reconsider properly citing this paper or another appropriate study.

• The introductory paragraph on Social Capital Theory and the section on Social Capital Theory is repetitive.

• Sato et al.’s (2019) study was conducted in Japan and based on Japanese experiences. Yet, it was used as a citation for the informal economic networks in Nigeria. Page 41, lines 20-22.

• The paper mentions the Sankofa philosophy in this sentence: “In the context of older Nigerians, the Sankofa philosophy highlights the value of honoring elders' experiences and knowledge while leveraging time-tested communal support systems.” What are the types of time-tested communal support systems?

• The authors should explain what traditional care models mean here: “It encourages societies to integrate traditional care models into modern frameworks, ensuring that cultural practices continue to sustain aging populations (Adam-Taylor et al., 2024; Asante, 2023).”

• The second recommendation, improving healthcare access for older adults in rural Nigeria, is not linked to social support. How can this support the formal social networks examined in the paper? The recommendation should focus on how healthcare access will enhance social support.

• “Training local residents, including retired healthcare workers, as community health volunteers can further bridge gaps, providing education and connecting older adults to formal healthcare systems.” Provide specific context on which gaps this sentence refers to and how connecting older adults to formal healthcare systems improves their social networks.

• Avoid redundancy by combining similar recommendations.

o The fourth and fifth recommendations are the same and should be collapsed into one.

o The sixth recommendation can also be collapsed into the third recommendation.

• For the seventh recommendation, the sentence “Research should also incorporate African perspectives on aging and community care, ensuring culturally relevant solutions. The authors should explain what ”African perspectives" mean here, tying that back to the study’s focus. Also, since the focus is on Nigeria, with a Table showing how the Nigerian system differs from other African countries such as South Africa, it makes better sense to argue for recognizing the Nigerian perspective. Africa is not monolithic.

• The paper lacks clear connection between the recommendations and the three aims of the paper..

• The conclusion introduces no new insights or critical reflections. Include a reflective discussion on the study’s limitations and areas for future research.

---

## [Editor Report]

Please may you address reviewer 2’s comment on including a reflective discussion on the study’s limitations and areas for future research.